# The Effect of Different Cadence on Paddling Gross Efficiency and Economy in Stand-Up Paddle Boarding

**DOI:** 10.3390/ijerph17134893

**Published:** 2020-07-07

**Authors:** Arkaitz Castañeda-Babarro, Jordan Santos-Concejero, Aitor Viribay, Borja Gutiérrez-Santamaría, Juan Mielgo-Ayuso

**Affiliations:** 1Health, Physical Activity and Sports Science Laboratory, Department of Physical Activity and Sports, Faculty of Psychology and Education, University of Deusto, 48007 Bizkaia, Spain; arkaitz.castaneda@deusto.es (A.C.-B.); borjagutierrez@deusto.es (B.G.-S.); 2Physiotherapy Department, Institute of Biomedicine (IBIOMED), University of Leon, Campus de Vegazana, 24071 Leon, Spain; 3Department of Physical Education and Sport, Faculty of Education and Sport, University of the Basque Country UPV/EHU, 01007 Vitoria-Gasteiz, Spain; jordan.santos@ehu.eus; 4Glut4Science, Physiology, Nutrition and Sport, 01004 Vitoria-Gasteiz, Spain; aitor@glut4science.com; 5Department of Biochemistry, Molecular Biology and Physiology, Faculty of Health Sciences, University of Valladolid, 42004 Soria, Spain

**Keywords:** economy, efficiency, stand up paddle, cadences, performance

## Abstract

*Background:* Due to the importance of energy efficiency and economy in endurance performance, it is important to know the influence of different paddling cadences on these variables in the stand-up paddleboarding (SUP). The purpose of this study was to determine the effect of paddling at different cadences on the energy efficiency, economy, and physiological variables of international SUP race competitors. *Methods:* Ten male paddlers (age 28.8 ± 11.0 years; height 175.4 ± 5.1 m; body mass 74.2 ± 9.4 kg) participating in international tests carried out two test sessions. In the first one, an incremental exercise test was conducted to assess maximal oxygen uptake and peak power output (PPO). On the second day, they underwent 3 trials of 8 min each at 75% of PPO reached in the first test session. Three cadences were carried out in different trials randomly assigned between 45–55 and 65 strokes-min^−1^ (spm). Heart rate (HR), blood lactate, perceived sense of exertion (RPE), gross efficiency, economy, and oxygen uptake (VO_2_) were measured in the middle (4-min) and the end (8-min) of each trial. *Results:* Economy (45.3 ± 5.7 KJ·l^−1^ at 45 spm vs. 38.1 ± 5.3 KJ·l^−1^ at 65 spm; *p* = 0.010) and gross efficiency (13.4 ± 2.3% at 45 spm vs. 11.0 ± 1.6% at 65 spm; *p* = 0.012) was higher during de 45 spm condition than 65 spm in the 8-min. Respiratory exchange ratio (RER) presented a lower value at 4-min than at 8-min in 55 spm (4-min, 0.950 ± 0.065 vs. 8-min, 0.964 ± 0.053) and 65 spm cadences (4-min, 0.951 ± 0.030 vs. 8-min, 0.992 ± 0.047; *p* < 0.05). VO_2_, HR, lactate, and RPE were lower (*p* < 0.05) at 45 spm (VO_2_, 34.4 ± 6.0 mL·kg^−1^·min^−1^; HR, 161.2 ± 16.4 beats·min^−1^; lactate, 3.5 ± 1.0 mmol·l^−1^; RPE, 6.0 ± 2.1) than at 55 spm (VO_2_, 38.6 ± 5.2 mL·kg^−1^·min^−1^; HR, 168.1 ± 15.1 beats·min^−1^; lactate, 4.2 ± 1.2 mmol·l^−1^; RPE, 6.9 ± 1.4) and 65 spm (VO_2_, 38.7 ± 5.9 mL·kg^−1^·min^−1^; HR, 170.7 ± 13.0 beats·min^−1^; 5.3 ± 1.8 mmol·l^−1^; RPE, 7.6 ± 1.4) at 8-min. Moreover, lactate and RPE at 65 spm was greater than 55 spm (*p* < 0.05) at 8-min. *Conclusion:* International male SUP paddlers were most efficient and economical when paddling at 45 spm vs. 55 or 65 spm, confirmed by lower RPE values, which may likely translate to faster paddling speed and greater endurance.

## 1. Introduction

Stand up paddleboarding (SUP) is a mixture of surfing and paddling [1], which has seen a significant increase in its number of practitioners. [2]. SUP involves propelling oneself with a long single-blade paddle using either side [3] on top of a board, which is longer, thicker, and wider than traditional surfboards [4]. There are three types of competitions in SUP: techniques, surfing, and racing [5]. While surfing competitions involve performing the highest quantity and quality of maneuvers in the waves, technical competitions and races entail performing a distance in the shortest time possible, both on open water and on flat water. In races, the most common distances to be covered are usually around 4–10 km (6.21 miles), although there are much longer distances [5].

There are different internal and external factors that have been studied in other surfing sports, which may be applicable to SUP [6]. Performance in these is conditioned by general weather conditions, which may be reflected in currents, wind, or waves. It is important that competitors know how to read the sea and take advantage of its waves so that they can cover several meters without having to make any effort. To do so, they must modify their trajectory and adjust it to each situation. Regarding internal variables, due to the importance of catching the waves mentioned above, and the speed stimuli needed during competitions, having a well-developed anaerobic metabolism can prove to be a determining factor.

On the other hand, SUP athletes are characterized by having good dynamic balance, as well as great strength in the trunk muscles [4]. This isometric strength of the trunk muscles, added to that of the gluteus and legs, is fundamental in counteracting the rotation forces that occur while paddling [7]. Furthermore, it seems that a well-trained aerobic metabolism is of great importance in different types of race [8]. Related to these major aerobic requirements, efficiency and economy are important aspects of endurance performance. Some studies claim that efficiency may be a better predictor of aerobic performance compared to maximal oxygen uptake (VO_2_ max) [9,10,11]. Conversely, other authors have shown that economy, defined as the energy demand at submaximal speeds, is one of the most discriminating factors of endurance performance, especially in athletes with similar VO_2_ max [12,13,14,15,16,17]. In this regard, Conley et al. [13] concluded that among highly trained and experienced runners of comparable capacity and similar VO_2_ max, running economy represents a large amount of the observed variation in performance over a 10 km run. While Mooses et al. [18] assert that economy is one of the factors explaining running performance, Bassett Jr et al. [19] claim that running economy and fractional utilization of VO_2_ max largely explains endurance performance.

Considering that this sport takes place in natural environments, with changing wind and waves, the paddling cadence during open sea SUP races may not be constant. Several studies in other sports have suggested that some cadences are more efficient than others [6,20,21,22,23,24,25,26]. In the case of cyclists and triathletes, Jacobs et al. [20] showed how lower cadences were more economical in trained subjects, while de Lucia et al. [21] obtained opposite results with world-class cyclists. In hand cycling, Kraaijenbrink et al. [25] obtained better efficiency values with low cadences than with high cadences, while Goosey et al. [26] comparing two crank lengths with two different cadences, obtained higher economy values with a short crank (180 mm) and higher cadence (85rev.min−1). With regard to running, Hafter et al. [23] after 6 weeks of training increased the cadence, although they managed to modify all the kinematic parameters, they did not manage to decrease the efficiency. Finally, with regard to traditional rowing, Aramendi et al. [24] recorded higher lactate and heart rates (HR) values at high cadences than at low cadences.

The question arises as to whether specific paddling cadences can be used, at 75% of PPO of the VO_2_ max, in order to improve efficiency and performance in SUP athletes. Thus, the main aim of this study was to analyze the influence of different paddling cadences: 45, 55, and 65 strokes-minute^−1^ (spm), on movement, gross efficiency and economy among elite competitive paddlers. Our hypothesis was that lower cadences would have better gross efficiency and economy compared to higher cadences.

## 2. Material and Methods

### 2.1. Participants

Ten male international SUP race competitors with at least four years’ competitive paddling experience were recruited for this study. Age, height, weight, anthropometric data, power test, and incremental test values of the participants are presented in Table 1. All participants were tested between 06/2019 and 08/2019, i.e., while they were during the competitive period of the 2019 season. The participants had undergone medical screening to ensure they were in good health and free from injury in the 12 months prior to research being conducted. Paddlers were fully informed of any risks associated with the experiments and provided with written informed consent to participate in the study, which was approved by the University of Deusto Ethics Committee (ref. ETK-13/18–19) in accordance with the latest version of the Declaration of Helsinki, Fortaleza (2013).

### 2.2. Experimental Trials

All participants underwent two test sessions with a week’s interval. On the first test day, an incremental exercise test was conducted to assess VO_2_ max and peak power output (PPO). These data were used to determine the intensity at which the participants should exercise in the second test session, which involved paddling at a constant sub-maximal intensity at 45, 55, and 65 spm to study how the cadence influenced paddling gross efficiency. The tests with different cadences involved three trials of 8 minutes’ (stage test) duration each, at 75% of PPO recorded in the VO_2_ max test. This intensity was selected to mimic the pace used during competitions [27]. The trials were 8 min long because it has been shown that this is a suitable duration for the purpose of evaluating gross efficiency and economy. [6,11]. There was a 10-min break between the trials. If during those 10 min the participant had not managed to descend from 100 beats·min^−1^, rest time was increased until they were able to do so. To prevent an order effect, the three cadence trials (45, 55, and 65 spm) were performed in a random sequence, and the paddling cadences selected for the study were based on peak cadence data obtained during a stress test by Schram et al. [4].

All paddle tests were performed on a modified ergometer (Ergo Vasa Swim, USA) [28] to ensure the same measurements in all tests. The VO_2_ max test was performed starting at 5 W, increasing by 5 W every minute and up to volitional exhaustion level [8]. All athletes were allowed to alternate paddling on each side ad libitum, and participants were given feedback about their cadence, visual feedback via the computer monitor of the ergometer, and sound feedback via metronome. Athletes were considered to have achieved maximal performance, and therefore reached their VO_2_ max, when at least two of the following criteria were fulfilled [29]: I) a plateau in VO_2_ max, defined as an increase of less than 1.5 mL·kg^−1^·min^−1^ in two consecutive workloads; II) RER >1.15; and III) maximal HR value (HRmax) >95% of the age-predicted maximum (220 – age). PPO (in W) was calculated as follows, taking every second into account (Equation (1)) [12]:(1)PPO=completed full intensity (W)+((second at final velocity/60 s)×5 W)

The ergometer was calibrated according to the manufacturer’s recommendations prior to all tests. To ensure a complete recovery and that there was no change in performance levels obtained in the tests, the measurements were taken at one-week intervals. Participants were also asked not to do any strenuous exercise 24–48 h prior to the assessments, and to eat a high-carbohydrate diet prior to the evaluation sessions. To avoid any variations in performance due to changes in the time of day at which the tests were performed, all evaluations were conducted at the same time of day.

### 2.3. Procedure

On the first day, height (cm) was obtained using a SECA 220 measuring rod (Hamburg, Germany), with precision to within 1 mm. Body Mass (BM; kg), percentage of body fat, body muscle (kg), and percentage of body muscle were measured using Inbody 770 (USA) within 0.1 kg. An incremental VASA ergometer test (Vasa, Inc., Essex Junction, VT, USA) was used to assess the VO_2_ max, and expired gases were collected and analyzed using a calibrated continuous breath-by-breath gas exchange and ventilation measurements at the mouth (Ergostik, Geratherm Respiratory GmbH, Bad Kissingen, Germany). The metabolic cart was calibrated to manufacturer recommendations before every test session.

On the second day, mean VO_2_ and power output (PO) were computed during the last 30 s in minute 4 (4-min) and minute 8 (8-min) of each trial (45–55 and 65 spm). Economy was calculated according to the Moseley and Jeukendrup equation (Equation (2)) [30] where the economy (KJ·L^−1^) equals the ratio between mean power output (PO) and mean steady-state oxygen uptake (L·min^−1^):(2)$$\mathrm{Economy}=\mathrm{PO}\;\left(W\right)/{\mathrm{VO}}_{2}\left({L\;\min}^{-1}\right)$$

Participants’ gross efficiency was assessed by calculating the amount of work completed relative to the amount of energy expended during each of the submaximal test stages, using the equation (Equation (3)):(3)$$\mathrm{Gross}\;\mathrm{Efficiency}\;\%=\left[\mathrm{work}\;\mathrm{rate}\left(W\right)/\mathrm{energy}\;\mathrm{cost}\;\left(J\;s^{-1}\right)\right]100$$

The energy consumed was calculated using the Weir equation [31] (Equation (4)):(4)$$Kcal=3.90\;VO_{2}\left(l\right)+1.10\;VCO_{2}\left(l\right)$$

Kcal·min−1 was converted to J·s^−1^ to quantify energy cost, and energy output as a percentage of energy cost was used to express efficiency. The RER results and the tables provided by Peronnet et al. [32] were used to calculate the percentage of oxidation of the fats of each paddler in each of the cadences.

Blood lactate measurements were taken before, in the middle (4-min), and at the end (8-min) of each trial of sub-maximal intensity (45–55 and 65 spm). Blood lactate was assessed via a Lactate Scout 2 handheld blood lactate analyzer (SensLab GmbH, Leipzig, Germany), while lactate measurements were taken from blood extracted from a finger, with the first drop always being discarded to avoid contamination. Likewise, a Polar HR monitor and transmitter (Polar Electro, Lake Success, NY, USA) was used to measure HR, which was recorded during the incremental test and throughout the submaximal workload trials.

The 10-point RPE scale [33] was used during the VO_2_ max test and at 4-min and 8-min of each submaximal trial (45–55 and 65 spm), and anyone other than researchers was forbidden from entering the laboratory in order for people’s presence not to influence participants’ RPE [34]. Verbal encouragement was given to all participants during the submaximal trials.

### 2.4. Statistical Analyses

Statistical data analyses were performed using the Statistical Package for the Social Sciences 24.0 (SPSS, Inc. Chicago, IL, USA), with descriptive statistics being calculated for each variable and expressed as mean ± standard deviation (SD), and range (min-max). The Shapiro–Wilk test (*n* <50) was conducted to determine the normality of the data, and the Levene test was used to check the uniformity of the variables analyzed. Differences between 4-min and 8-min tests for each variable in each cadence condition were assessed via a dependent t-test. VO_2_, blood lactate, RPE, HR, economy, gross efficiency, and RER were compared across different cadences using one-way ANOVA with the cadences as the fixed factor. Bonferroni post-hoc test was applied for pairwise comparisons among groups. Additionally, effect sizes were calculated using partial eta square and η^2^p, although because this measure was likely to overestimate said effect sizes, values were interpreted according to that indicating that there was no effect if 0 ≤ η^2^p < 0.05; minimum effect if 0.05 ≤ η^2^p < 0.26; moderate effect if 0.26 ≤ η^2^p < 0.64; and a strong effect if η^2^p ≥ 0.64 [35]. Statistical significance for all analyses was set at *p* < 0.05.

## 3. Results

Table 2 shows the data obtained in economy, gross efficiency, and RER. In contrast with RER, both economy and gross efficiency evidenced differences among cadences in the 8-min test (*p* < 0.05). More specifically, in 8-min, 45 spm showed a lower economy and gross efficiency than the 65 spm (*p* < 0.05). Additionally, in contrast with RER, which presented a higher value at 8-min than 4-min in 55 spm and 65 spm cadences (*p* < 0.05), there was no difference in economy and gross efficiency between 4-min and 8-min in any of the 3 cadences (*p* > 0.05).

Table 3 depicts the results in VO_2_, HR, Lactate, and RPE of each cadence. VO_2_, HR, lactate, and RPE evidenced differences among cadences in the 8-min test (*p* < 0.05). The cadence of 65 spm presented a higher VO_2_, HR, lactate, and RPE than 45 spm in the 8-min test (*p* < 0.05). Moreover, lactate and RPE in 65 spm presented higher value than 55 spm in the 8-min test (*p* < 0.05). On the other hand, HR and RPE was higher at 8-min than 4-min in 55 spm and 65 spm. However, at 8-min lactate presented a higher value than at 4-min in 65 spm (*p* < 0.05).

## 4. Discussion

The main aim of this study was to determine the influence of cadence (45, 55, and 65 spm) on gross efficiency, economy, HR, blood lactate, RPE, and RER in an 8-min test in elite competitive male paddlers. The main finding of this study was that paddling gross efficiency and economy were higher at 45 spm than at 65 spm, and RER was lower at 45 spm. HR and lactate were lower during the 45 spm trial and likely reflected the lower VO_2_ and higher gross efficiency associated with this cadence. Movement efficiency, combined with maximum aerobic power and anaerobic threshold, are three physiological measures that together can be used to predict performance in endurance sports [10,11,21,30,36]. This efficiency of movement is influenced by the energy substrate used and the percentage of slow (more efficient) muscle fibers [37,38]. Since SUP is an endurance sport, energy-saving and, therefore, efficiency is an important factor in racing performance.

There are no other studies analyzing the influence of different stroke cadences on movement gross efficiency and economy in SUP. We found that paddling at 45 spm was 13% and 17.5% more efficient and 12.9% and 16.9% more economical than at 55 and 65 spm, respectively. The upper body is generally accepted as having a greater proportion of fast-twitch fibers when compared to the lower body [39,40]. Interestingly, the energetically optimal cadence has been reported to be higher in a model with more fast-twitch fibers than a model with more slow-twitch fibers [41], consistent with predictions from the literature [42,43,44]. This would partially explain why higher cadences displayed worse gross efficiency values in the current study. Similarly, higher cadences may be related to greater instability, which would imply higher needs of muscle activation for postural control and would ultimately lead to higher energy consumption, and therefore, worse efficiency [45]. When compared to studies in other sports, our findings were in agreement with those of Neilsen et al. [46] and Jacobs et al. [20] in cycling, Gonzalez-Aramendi [24] in traditional rowing, and Kraaijenbrink et al. [25] in hand cycling, as all of them found that lower cadences were related to better efficiency and economy values. In contrast, Lucia et al. [21] and Mora-Rodriguez and Aguado-Jimenez [47] in cycling, or Goosey et al. [26] in hand cycling reported that higher cadences were more efficient and economical. These differences may be due to the competitive level of the sport according to the number of practitioners who perform the activity, the kind of exercise performed (upper vs. lower body), and the different protocol used to determinate the movement efficiency/economy (i.e., incremental vs. constant intensity test).

In this regard, SUP is a relatively new sport and has a low level of professionalization when compared to other types of sport [27]. It can therefore be assumed that well-trained cyclists [47], world-class cyclists [21], or top-level athletes [26] have more training experience in their disciplines than SUP athletes (5.8 ± 1.9 years). Longer training experience in athletes of consolidated disciplines may imply a greater volume of training over the years, which is known to produce various physiological adaptations related to efficiency and economy. Similarly, neuromotor recruitment can be improved by the greater volume of training on the part of experienced athletes [11]. Moreover, higher cadences appear to affect negatively the force effectiveness [48], which is a measure of the paddling technique. Higher cadences are believed to increase internal kinetic energy fluctuations (rotation of the extremities). Even if this energy flow can also be used as external work [49], it is likely associated with an increased oxygen cost and, therefore, it would affect efficiency and economy negatively. Higher cadences may also affect the inertial components of the paddling forces, which are related to the kinetic energy fluctuations, similar to what has been observed in cycling [50].

Another aspect to take into account in relation to efficiency and economy is the number of muscles involved in the specific activity, diffusion area, and diffusion distance [51,52,53]. Since the upper body is believed to have a greater proportion of fast-twitch fibers [39,40], it may explain the slower kinetics of VO_2_ and cardiac output in the transition from rest to exercise intensities of between 30–90% PPO in the upper body when compared to the lower body [39,54]. Lastly, the evaluation protocol used is another factor that should be considered when monitoring efficiency and economy. The duration of the exercise bouts during the examination may affect values, as protocols with the longer bouts [20] report a continuous decrease in efficiency in terms of time and cadence, and, in addition, the use of a minute 4 ramp protocol [25,26,47] may underestimate steady-state VO_2_.

Interestingly, we found that the substrate utilization varied among cadences. When paddling at 45 spm the percentage of fat used to obtain energy was 21.4% compared to 11.6% at 55 spm and 2% at 65 spm [32]. This means consuming twice (55 spm) and ten times (65 spm) as much fat, respectively (Table 2). Considering that most races in the best SUP circuits in the world take about one hour, these differences in substrate utilization may imply that an efficient paddler will end the race with a higher amount of carbohydrates available [55,56]. This advantage in terms of availability of substrates may be of great importance in finishing a competition at high intensity [57]. The results obtained from our study reflect lower RER (greater fat utilization) with the 45 spm cadence, which would allow paddlers to save on their most valuable energy substrate.

It is important to take into account that athletes have a naturally chosen cadence, which is usually the most economical [58]. This subconscious fine-tuning of movement biomechanics is referred to as self-optimization, which appears to be a physiological adaptation resulting from greater training experience [59]. In order to optimally recruit motor units, cadences may require specific training. This lack of familiarity with some cadences has been studied in cycling [60,61], and the results indicate that the most used cadences, compared to the unknown ones, are the ones that obtain the best results. However, this type of study with different cadences avoids the possible motor learning that may occur when exercising with several cadences. To allow for any possible motor learning effect on movement gross efficiency, it may be useful to add training for a period of time to the design of this type of study before evaluating gross efficiency [23].

We have to acknowledge several limitations. It is important to highlight the small sample size in this study. However, it is also true that it is very difficult to obtain larger samples in elite sports and in this case, in a new sport with a low level of professionalization [27]. Moreover, this study used a work rate of 75% of peak power, and so our results may not be applicable to higher or lower work rates. Likewise, we should take into account the cadence naturally selected by the paddlers, because as we have commented previously, it can influence the economy results. Therefore, individual athletes should try various cadences and drags to personally optimize economy. Since SUP racing is not performed at a constant work rate, we suspect that optimal cadence may vary depending on waves, wind, and other environmental conditions; having said this, the use of 45 spm may be an effective cadence with which to compare other paddling speeds.

This study contributes to knowledge about paddlers’ physiological response, besides also contributing to the good design of both race and training strategies. The limitations of extrapolating these data so as to put them into practice in the field are evident, as the differences between simulated laboratory practice and practice in the field in SUP mode have not yet been studied. One study made the same comparison for similar sports, such as kayaking, and concluded that the ergometer accurately simulates the physiological demands of kayaking (50). Future studies should contrast the differences between laboratory and field measurements in SUP.

## 5. Conclusions

International male SUP paddlers were most gross efficiency and economical when paddling at 45 spm vs. 55 or 65 spm, as confirmed by lower RPE values in measurements made in a laboratory. Similarly, this gross efficiency and economy shown at 45 spm implied a greater use of fat as an energy substrate. Those improvements may likely translate to faster paddling speed and greater endurance, and they may be helpful to coaches and athletes in determining optimal GE and economy, as these differences in competition are likely to yield meaningful improvement in performance.

### Practical Applications

This study attempted to determine whether a low vs. high cadence would give rise to better energy gross efficiency and economy in paddlers participating in international competitions. The 45-stroke cadence was the most gross efficiency and economical one, in addition to obtaining lower psycho-physiological values than the other cadences. These results reinforce the idea that paddling at a low cadence can maximize sustainable energy production while minimizing metabolic stress. These paddlers could improve their performance in long-distance tests by paddling at a low cadence vs. a high cadence.

## Figures and Tables

**Table 1 ijerph-17-04893-t001:** Participants’ age, body composition, and athletic performance characteristics (*n* = 10).

Variable	Mean ± SD	Range (Min–Max)
Age (y)	28.8 ± 11.0	18.0–46.0
Height (cm)	175.4 ± 5.1	167.0–186.5
Body mass (kg)	74.2 ± 9.4	61.3–92.9
Muscular mass (kg)	36.7 ± 3.9	30.5–44.0
Muscular mass (%)	49.6 ± 2.4	45.9–55.1
Body fat (%)	12.7 ± 3.9	6.7–17.9
PPO in 10 s (W)	336.7 ± 88.7	210.0–528.0
VO_2_ max (mL·kg^−1^·min^−1^)	49.9 ± 3.7	45.2–57.8
HR max (beats·min^−1^)	183.2 ± 14.1	164–207
PPO at VO_2_ max (W)	160.0 ± 19.5	120–190
Relative PPO (W·kg^−1^)	2.2 ± 0.3	1.7–2.8
Muscle relative PPO (W·kg^−1^)	4.4 ± 0.6	3.5–5.6

Data are expressed as mean ± standard deviation. PPO, peak power output; HRmax, maximal heart rate; VO2 max, maximum oxygen uptake; PO, power output.

**Table 2 ijerph-17-04893-t002:** Economy, gross efficiency, and RER variables of SUP Paddlers at 4-min and 8-min.

	45 spm	55 spm	65 spm	*p*	η^2^p
**Economy (KJ·l^−1^)**					
4-min	42.8 ± 6.0	40.5 ± 9.4	40.4 ± 6.7	0.238	0.150
8-min	45.3 ± 5.7	39.9 ± 7.7	38.1 ± 5.3 ^a^	0.010	0.436
**Gross Efficiency (%)**					
4-min	12.7 ± 2.2	11.9 ± 2.9	11.9 ± 2.2	0.166	0.192
8-min	13.4 ± 2.3	11.6 ± 2.4	11.0 ± 1.6 ^a^	0.012	0.430
**RER**					
4-min	0.918 ± 0.05	0.950 ± 0.065 *	0.951 ± 0.030 *	0.206	0.187
8-min	0.934 ± 0.04	0.964 ± 0.053	0.992 ± 0.047	0.081	0.280

Data are shown as mean ± standard deviation. 4-min and 8-min refer to the middle and the end of the test, respectively; *p*, Significant differences between cadences by one factor ANOVA (cadences); ^a^, significant differences regarding 45 spm using the Bonferroni test; *, significant differences between 4-min and 8-min using dependent *t*-test.

**Table 3 ijerph-17-04893-t003:** VO_2_, HR, Lactate, and RPE variables of SUP Paddlers at 4-min and 8-min.

	45 spm	55 spm	65 spm	*p*	η^2^p
**VO_2_ (mL·kg^−1^·min^−1^)**					
4-min	35.0 ± 5.1	37.3 ± 6.5	37.3 ± 5.9	0.224	0.158
8-min	34.4 ± 6.0	38.6 ± 5.2^a^	38.7 ± 5.9^a^	0.020	0.415
**HR (beats·min^−1^)**					
4-min	157.4 ± 16.6 *	159.8 ± 19.7 *	164.2 ± 13.4 *	0.074	0.252
8-min	161.2 ± 16.4	168.1 ± 15.1 ^a^	170.7 ± 13.0 ^a^	0.007	0.463
**Lactate (mmol L^−1^)**					
4-min	3.4 ± 1.0	3.9 ± 1.5	4.1 ± 1.0 *	0.171	0.192
8-min	3.5 ± 1.0	4.2 ± 1.2 ^a^	5.3 ± 1.8 ^a,b^	0.006	0.506
**RPE**					
4-min	6.0 ± 2.1	6.2 ± 1.5 *	6.4 ± 1.8 *	0.461	0.077
8-min	6.0 ± 1.7	6.9 ± 1.4	7.6 ± 1.4 ^a,b^	<0.001	0.618

Data are shown as mean ± standard deviation. 4-min and 8-min refer to the middle and the end of the test, respectively. *p*: Significant differences between cadences by one-factor ANOVA (cadences); ^a^, Significant differences regarding 45 spm using the Bonferroni test; ^b^, significant differences regarding 55 spm using the Bonferroni test; *, Significant differences between 4-min and 8-min using dependent *t*-test.

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
