# Peer review of "The Effect of Different Cadence on Paddling Gross Efficiency and Economy in Stand-Up Paddle Boarding"

_ijerph, 2020, doi:10.3390/ijerph17134893_

Round 1

Reviewer 1 Report

This paper reports difference in metabolic responses between cadences for SUP. This paper is mostly well written, but the authors should correct the points listed below.

Major points

The authors need to discuss about physiological reason why gross efficiency and economy were higher in lower cadence.

Minor points

L143. What is VO2 max peak?

L185, L210, L212, Table 3. Please correct from VO2max to VO2. This is just a VO2 data, not maximal value.

Table 2. Please revise number of significant digits for RER at 45spm.

Author Response

Point-by-Point Response to Reviewer’s Comments

We would like to sincerely thank the reviewers for their helpful recommendations. We have seriously considered all the comments and carefully revised the manuscript accordingly. Revisions are highlighted in yellow through the manuscript to indicate where changes have taken place. We feel that the quality of the manuscript has been significantly improved with these modifications and improvements based on the reviewers’ suggestions and comments. We hope our revision will lead to an acceptance of our manuscript for publication in International Journal of Environmental Research and Public Health.

In advance,

Kind regards

REVIEWER 1

This paper reports difference in metabolic responses between cadences for SUP. This paper is mostly well written, but the authors should correct the points listed below.

Major points

REVIEWER: The authors need to discuss about physiological reason why gross efficiency and economy were higher in lower cadence.

AUTHORS: Thank you for your observation. An additional paragraph has been added to the discussion with the aim of improving it.

“Another possible explanation is that higher cadences appear to affect negatively the force effectiveness [50], which is measure of the paddling technique. Higher cadences are believed to increase internal kinetic energy fluctuations (rotation of the extremities). Even if this energy flow can also be used as external work [51], it is likely associated with an increased oxygen cost and, therefore, it would affect efficiency and economy negatively. Higher cadences may also affect the inertial components of the paddling forces, which are related to the kinetic energy fluctuations, similar to what has been observed in cycling [52].”

Minor points

REVIEWER: L143. What is VO2 max peak?

AUTHORS: Thank you for your observation. “peak” word has been removed from the text.

REVIEWER: L185, L210, L212, Table 3. Please correct from VO2max to VO2. This is just a VO2 data, not maximal value.

AUTHORS: Thank you for your observation. Mistakes in the text have been corrected.

REVIEWER: Table 2. Please revise number of significant digits for RER at 45spm.

AUTHORS: Thank you for your observation. The missing numbers have been added.

Reviewer 2 Report

Current research entitled "The effect of different cadence on paddling gross efficiency and economy in stand-up paddle boarding" is very interesting for the scientific ad technical community. The authors have done a good job, however, there are some minor points to consider. I recommend the authors pay special attention to the comment made on Line 280 of the Discussion section.

Minor specific comments

Abstract
Line 20 to 22: The argument is not enough. Rephrase.
Line 25: Add the units
Line 29-30. “Each...min-1.” rephrase.
Line 32: Add numbers.
Line 33: Withdraw “significant” along with the manuscript
Line31: Withdraw “The major finding...significantly”... Withdraw significantly along with the manuscript.

Introduction
Line 69: [12-21] ? Is it correct? If so, it is important to evidence the importance of each study in the Introduction, instead of just condensate to many studies in the same sentence.
Line 77: (188mm). Consider adding a space between the number and units along with the manuscript.

Materials and Methods
Line 101: Consider standardize “VO2max” quotation along with the manuscript.
Line 103-105: Rewrite and consider merging with other paragraphs. Paragraph length should be attempted. Line: 139-141- The anthropometrical procedures should be presented in the right order it was collected. “On the first day...”
Line: 155-156: ..using Eq III withdrawn following...
Line: 188: .. partial eta square ().

Results
Line 200-202: 4 minute and 8 minute...standardize along with the manuscript. Line 205: ...gross efficiency and RER of SUP...
Line 208: 45 spm ... standardize along with the manuscript.
Line 195-204; 201-216: It is not necessary to reply to the Table results.
Line 217: Table 3 title rewrite as Table 2.

Discussion
Line 224: withdraw various
Line 230: rewrite to VO2max...as authors have used along with the manuscript
Paragraph 1 and 2 should be merged.
Line 251: ..;(upper vs. lower body)...the same for Line 253
Line 280: Is this sentence necessary? It does not sound good to hear that elite athletes were not familiarized with the cadences. It is quite complicated for your results and should be carefully discussed. Authors should also present this limitation in the Methods, and how they have minimized such crucial design limitation. Line 312: Authors should follow the same structure of numbers along with the manuscript. See 2. And 2.1.

Author Response

Point-by-Point Response to Reviewer’s Comments

We would like to sincerely thank the reviewers for their helpful recommendations. We have seriously considered all the comments and carefully revised the manuscript accordingly. Revisions are highlighted in yellow through the manuscript to indicate where changes have taken place. We feel that the quality of the manuscript has been significantly improved with these modifications and improvements based on the reviewers’ suggestions and comments. We hope our revision will lead to an acceptance of our manuscript for publication in International Journal of Environmental Research and Public Health.

In advance,

Kind regards

REVIEWER 2

REVIEWER: Current research entitled "The effect of different cadence on paddling gross efficiency and economy in stand-up paddle boarding" is very interesting for the scientific ad technical community. The authors have done a good job, however, there are some minor points to consider. I recommend the authors pay special attention to the comment made on Line 280 of the Discussion section.

Minor specific comments

Abstract

REVIEWER: Line 20 to 22: The argument is not enough. Rephrase.

AUTHORS: Thank you for your recommendation. The sentence in the text has been modified. “Due to the importance of energy efficiency and economy in endurance performance, it is important to know the influence of different paddling cadences on these variables in the stand-up paddleboarding (SUP).”

REVIEWER: Line 25: Add the units

AUTHORS: Thank you for your observation. The units have been added.

REVIEWER: Line 29-30. “Each...min-1.” rephrase.

AUTHORS: Thank you for your observation. The sentence in the text has been modified by including the following sentence: “Three cadences were carried out in different trials randomly assigned between 45-55 and 65 strokes-min-1

REVIEWER: Line 32: Add numbers.

AUTHORS: Thank you for your observation. The authors have added number in the abstract´s results section: Economy (45.3±5.7 KJ·l-1 at 45 spm Vs 38.1±5.3 KJ·l-1 at 65 spm; p=0.010) and gross efficiency (13.4±2.3% at 45 spm Vs 11.0±1.6% at 65 spm; p= 0.012) was higher during de 45 spm condition than 65 spm in the 8-min. Respiratory exchange ratio (RER) presented a lower value at 4-min than at 8-min in 55 spm (4-min: 0.950±0.065 vs. 8-min: 0.964±0.053) and 65 spm cadences (4-min: 0.951±0.030 vs. 8-min: 0.992±0.047) (p<0.05). VO2 max, HR, lactate and RPE were lower (p<0.05) at 45 spm (VO2 max: 34.4 ± 6.0 ml·kg-1·min-1; HR: 161.2 ± 16.4 beats·min-1; lactate: 3.5 ± 1.0 mmol·l-1; RPE: 6.0 ± 2.1) than at 55 spm  (VO2 max: 38.6 ± 5.2 ml·kg-1·min-1; HR: 168.1 ± 15.1 beats·min-1; lactate: 4.2 ± 1.2 mmol·l-1; RPE: 6.9 ± 1.4) and 65 spm spm  (VO2 max: 38.7 ± 5.9 ml·kg-1·min-1; HR: 170.7 ± 13.0 beats·min-1; 5.3 ± 1.8 mmol·l-1;RPE: 7.6 ± 1.4) at 8-min. Moreover, lactate and RPE at 65 spm was greater than 55 spm (p<0.05) at 8-min.

REVIEWER: Line 33: Withdraw “significant” along with the manuscript. Line31: Withdraw “The major finding...significantly”... Withdraw significantly along with the manuscript.

AUTHORS: Thank you for your observation. The word “significant” and “significantly” have been removed from the manuscript (except for the tables).

Introduction

REVIEWER: Line 69: [12-21] ? Is it correct? If so, it is important to evidence the importance of each study in the Introduction, instead of just condensate to many studies in the same sentence.

AUTHORS: Thank you for your comment. The authors have added some sentences explain the evidence of different studies: “  In this regard, Conley et al. [13] concluded that among highly trained and experienced runners of comparable capacity and similar VO2max, running economy represents a large amount of the observed variation in performance over a 10 km run. While Mooses et al. [18] assert that economy is one of the factors explaining running performance, Bassett Jr et al. [19] claim that running economy and fractional utilization of VO2max largely explain endurance performance.”

REVIEWER: Line 77: (188mm). Consider adding a space between the number and units along with the manuscript.

AUTHORS: Thank you for your observation. A space has been added between the number and the corresponding unit.

Materials and Methods

REVIEWER: Line 101: Consider standardize “VO2max” quotation along with the manuscript.

AUTHORS: Thank you for your observation. A VO2max test was performed on the first day of assessment, and the values shown in the table refer to this data. However, the measurements made on the second day have been called VO2 because the maximum is not reached during trials at different cadences.

REVIEWER: Line 103-105: Rewrite and consider merging with other paragraphs. Paragraph length should be attempted.

AUTHORS: Thank you for your observation. Following your recommendations, the paragraph has been removed from the text.

REVIEWER: Line: 139-141- The anthropometrical procedures should be presented in the right order it was collected. “On the first day...”.

AUTHORS: Thank you for your observation. Temporary references were added in the text: “On the first day…” and “On the second day…”.

REVIEWER: Line: 155-156: ..using Eq III withdrawn following...

AUTHORS: Thank you for your observation. Following… has been withdrawn.

REVIEWER: Line: 188: .. partial eta square ().

AUTHORS: Thank you for your observation. The authors have added “partial eta square” in line 188.

Results

REVIEWER: Line 200-202: 4 minute and 8 minute...standardize along with the manuscript. Line 205: ...gross efficiency and RER of SUP...

AUTHORS: Thank you for your observation. The way of writing has been homogenized with respect to the data obtained in minute 4 and minute 8 of the test, we have use 4-min and 8-min to standardized these terms.

REVIEWER: Line 208: 45 spm ... standardize along with the manuscript.

AUTHORS: Thank you for your observation. The nomenclatures have been homogenized.

REVIEWER: Line 195-204; 201-216: It is not necessary to reply to the Table results.

AUTHORS: Thank you for your observation. The added text explains the results but does not repeat the data presented in the tables, so we believe that they can add value to the article.

REVIEWER: Line 217: Table 3 title rewrite as Table 2.

AUTHORS: Thank you for your observation. The title has been rewritten as table number 2: “VO2, HR, Lactate and RPE variables of SUP Paddlers at minutes 4 and 8”.

Discussion

REVIEWER: Line 224: withdraw various

AUTHORS: Thank you for your observation. The word “various” has been removed from de text.

REVIEWER: Line 230: rewrite to VO2max...as authors have used along with the manuscript

AUTHORS: Thank you for your observation. The authors have rewritten VO2max along with the manuscript

REVIEWER: Paragraph 1 and 2 should be merged.

AUTHORS: Thank you for your observation. Paragraphs 1 and 2 of the discussion have been merged.

REVIEWER: Line 251: ..;(upper vs. lower body)...the same for Line 253

AUTHORS: Thank you for your observation. The suggestions made have been corrected.

REVIEWER: Line 280: Is this sentence necessary? It does not sound good to hear that elite athletes were not familiarized with the cadences. It is quite complicated for your results and should be carefully discussed. Authors should also present this limitation in the Methods, and how they have minimized such crucial design limitation.

AUTHORS: Thank you for your observation. The authors have added a sentence discussing this aspect: It is important to take into account that athletes have a naturally chosen cadence, which is usually the most economical [58]. This subconscious fine-tuning of movement biomechanics is referred to as self-optimization, which appears to be a physiological adaptation resulting from greater training experience [59].”

A sentence has been added in the limitations: “Likewise, we should take into account the cadence naturally selected by the paddlers, because as we have commented in the discussion, it can influence the economy results.”

REVIEWER: Line 312: Authors should follow the same structure of numbers along with the manuscript. See 2. And 2.1.

AUTHORS: Thank you for your observation. The section on "practical applications" has been included as point 6 of the article.

Round 2

Reviewer 1 Report

Thank you for adjusting the manuscript to the recommendations given.

Minor comments.

In some sentence, revising from VO2 max to VO2 was not completed; for example, abstract, L211, and L270. Please revise it.

L31, 32. Please revise from “Vs” to “vs”.

Table 3. Lactate. -1 should be superscript.

L195. η2p. 2 should be superscript.

Reference

Ref 49. Title, eYciency?

Author Response

Point-by-Point Response to Reviewer’s Comments

We would like to sincerely thank the reviewer again for his/her helpful recommendations. We have seriously considered all the comments and carefully revised the manuscript accordingly. Revisions are highlighted in yellow through the manuscript to indicate where changes have taken place. We feel that the quality of the manuscript has been significantly improved with these modifications and improvements based on the reviewers’ suggestions and comments. We hope our revision will lead to an acceptance of our manuscript for publication in International Journal of Environmental Research and Public Health.

In advance,

Kind regards

REVIEWER 1

Thank you for adjusting the manuscript to the recommendations given.

Minor Comments

REVIEWER: In some sentence, revising from VOmax to VO2 was not completed; for example, abstract, L211, and L270. Please revise it.

AUTHORS: Thank you for your observation. The authors have corrected these mistakes.

REVIEWER: L31, 32. Please revise from “Vs” to “vs”.

AUTHORS: Thank you for your observation. The authors have modified Vs to vs.

REVIEWER: Table 3. Lactate. -1 should be superscript.

AUTHORS: Thank you for your observation. The authors have included -1 in superscript.

REVIEWER: L195. η2p. 2 should be superscript.

AUTHORS: Thank you for your observation. The authors have included 2 in superscript.

REVIEWER: Reference, Ref 49. Title, eYciency?

AUTHORS: Thank you for your observation. The authors have modified this mistake.

This manuscript is a resubmission of an earlier submission. The following is a list of the peer review reports and author responses from that submission.

Round 1

Reviewer 1 Report

General comments 

current research entitled “the effect of different cadence in paddling gross efficiency and economy in stand up paddle boarding” is very interesting for scientific and technical community. The paper has originality, rationality and completeness in the first reading. For better reading flow, variables should be rather presented in the same order from the abstract To the conclusion. Authors should make an additional effort for manuscript quality improvement and publication, following the major points presented below.

Abstract

line 20 To 22 : the absence of studies in some topic is not a reason for a new research. Authors should rewrite and clarify the need for this study knowing the literature background.

line 29-30: add the efficiency and economy variables

line 31: withdraw the major finding significantly. Withdraw significantly along with the manuscript.

line 35: vo2 abbreviation was not presented.

line 36: p=0.000? Rewrite using <.

Introduction

line 48-49: rewrite the sentence and add the three types of...

line 52: consider a space between the number and the unity

line 65-69: This paragraph is too short compared to the previous. Rewrite and balance the paragraphs length along with the manuscript.

line 72-74:the argument seems quite obvious. Authors should consider reviewing the manuscript arguments To present the main question.

line 74: Keep the standardised references quotation.

line 75: specific paddling cadences can be used in order to. Specify which intensity

line 77: is there any hypothesis?

Material and Methods

line 80: table 1 measures should be better explained before its appearance. Table 1 titled should be detailed presented

line 93-95: consider standardising vo2max abbreviation along with the manuscript

line 150: analyzer (2x)

line 154-154: a paragraph with two lines...consider reviewing.

line 161: descriptive statistics

line 162: specify range

line 167: while a bonferroni, withdraw while

Results

line 176: Why have the authors presented ge and efficiency?it ia difficult to follow the manuscript reading using both terms.

line 180: it is not necessary repeating table results in the text

line 176: withdraw ge

line 197: it is not necessary repeating results from the table

line 212: why have the authors opened another paragraph? Keep variables results in the same order from the abstract.

Discussion

line 235:withdraw to the best of our knowledge

line 252-253: conversion of type 2 to  ... authors have not assessed this data. Withdraw

line 285: to allow for...

Author Response

Point-by-Point Response to Reviewer’s Comments

We would like to sincerely thank the reviewers for their helpful recommendations. We have seriously considered all the comments and carefully revised the manuscript accordingly. Revisions are highlighted in yellow through the manuscript to indicate where changes have taken place. We feel that the quality of the manuscript has been significantly improved with these modifications and improvements based on the reviewers’ suggestions and comments. We hope our revision will lead to an acceptance of our manuscript for publication in International Journal of Environmental Research and Public Health.

In advance,

King regards

REVIEWER 1

General comments 

current research entitled “the effect of different cadence in paddling gross efficiency and economy in stand up paddle boarding” is very interesting for scientific and technical community. The paper has originality, rationality and completeness in the first reading. For better reading flow, variables should be rather presented in the same order from the abstract To the conclusion. Authors should make an additional effort for manuscript quality improvement and publication, following the major points presented below.

Abstract

REVIEWER: line 20 To 22 : the absence of studies in some topic is not a reason for a new research. Authors should rewrite and clarify the need for this study knowing the literature background.

AUTHORS: Thank you for your recommendation. We have modified the sentence trying to give an explanation of why this research is necessary: “Although the impact of the use of different cadences on energy efficiency and economy has been studied in some cyclic sports and the relationship between these variables and sports performance is known, there are no studies on this subject in stand up paddleboarding (SUP).”

REVIEWER: line 29-30: add the efficiency and economy variables

AUTHORS: Thank you for your recommendation. We have added the words efficiency and economy in line 30.

REVIEWER: line 31: withdraw the major finding significantly. Withdraw significantly along with the manuscript.

AUTHORS: Thank you for your recommendation. We have corrected the contribution

REVIEWER: line 35: vo2 abbreviation was not presented.

AUTHORS: Thank you for your observation. We have changed the VO2 by VO2 max previously defined (maximal oxygen uptake (VO2 max)).

REVIEWER: line 36: p=0.000? Rewrite using <.

AUTHORS: Thank you for your recommendation. We have changed p=0.000 by p<0.001.

Introduction

REVIEWER: line 48-49: rewrite the sentence and add the three types of...

AUTHORS: Thank you for your recommendation. We have rewritten the sentence to clarify the three types of SUP: “There are three types of competitions in SUP: techniques, surfing and racing [5].”

REVIEWER: line 52: consider a space between the number and the unity

AUTHORS: Thank you for your recommendation. We have added space between 4-10 and km.

REVIEWER: line 65-69: This paragraph is too short compared to the previous. Rewrite and balance the paragraphs length along with the manuscript.

AUTHORS: Thank you for your observation. Contribution solved.

REVIEWER: line 72-74: the argument seems quite obvious. Authors should consider reviewing the manuscript arguments To present the main question.

AUTHORS: Thank you for your recommendation. The authors have added a paragraph in the introduction.In the case of cyclists and triathletes, Jacobs et al. [22] showed how lower cadences were more economical in trained subjects, while de Lucia et al. [23] obtained opposite results with world-class cyclists. In hand cycling, Kraaijenbrink et al. [27] obtained better efficiency values with low cadences than with high cadences, while Goosey et al. [28] comparing two crank lengths with two different cadences, obtained higher economy values with a short crank (180mm) and higher cadence (85rev.min-1). With regard to the run, Hafter et al. [25] after 6 weeks of training for increase the cadence, although they managed to modify all the kinematic parameters, they did not manage to decrease the efficiency. Finally, with regard to traditional rowing, Aramendi et al. [26] recorded higher lactate and heart rates (HR) values at high cadences than at low cadences.”

REVIEWER: line 74: Keep the standardised references quotation.

AUTHORS: Thank you for your observation. We have standardised the reference.

REVIEWER: line 75: specific paddling cadences can be used in order to. Specify which intensity

AUTHORS: Thank you for your observation. We have specified the intensity in the text: “at 75% of PPO of the VO2 max”

REVIEWER: line 77: is there any hypothesis?

AUTHORS: Thank you for your observation. We have a hypothesis in the text: “Our hypothesis was that lower cadences would have better gross efficiency and economy compared to higher cadences.”

Material and Methods

REVIEWER: line 80: table 1 measures should be better explained before its appearance. Table 1 titled should be detailed presented

AUTHORS: Thank you for your recommendation. In order to avoid being redundant, the authors have indicated in the text the data that can be found in the table: “Age, height, weight, anthropometric data, power test and incremental test values of the participants is presented in Table 1.”

REVIEWER: line 93-95: consider standardising vo2max abbreviation along with the manuscript

AUTHORS: Thank you for your recommendation. The authors have standardized VO2 max abbreviation along with the manuscript.

REVIEWER: line 150: analyzer (2x)

AUTHORS: Thank you for your observation. The authors have rewritten those lines to clarify this concern: “Blood lactate was assessed via a Lactate Scout 2 handheld blood lactate analyzer (SensLab GmbH, Leipzig, Germany)”

REVIEWER: line 154-154: a paragraph with two lines...consider reviewing.

AUTHORS: Thank you for your recommendation. The authors have linked this paragraph with previous paragraph.

REVIEWER: line 161: descriptive statistics

AUTHORS: Thank you. The concerns have been solved.

REVIEWER: line 162: specify range

AUTHORS: Thank you for your interest. The authors have added the range “(min-max).”

REVIEWER: line 167: while a bonferroni, withdraw while

AUTHORS: Thank you for your recommendation. The authors have deleted while.

Results

REVIEWER: line 176: Why have the authors presented ge and efficiency?it ia difficult to follow the manuscript reading using both terms.

AUTHORS: Thank you for your recommendation. The authors have changed GE by gross efficiency along the manuscript.

REVIEWER: line 180: it is not necessary repeating table results in the text

AUTHORS: Thank you for your recommendation. The authors have deleted repeating table results in the text.

REVIEWER: line 176: withdraw ge

AUTHORS: Thank you for your recommendation. We have deleted it.

REVIEWER: line 197: it is not necessary repeating results from the table

AUTHORS: Thank you for your recommendation. The authors have deleted repeating table results in the text.

REVIEWER: line 212: why have the authors opened another paragraph? Keep variables results in the same order from the abstract.

AUTHORS: Thank you for your recommendation. We have rewritten that paragraph: “Table 3 depicts the results in VO2 max, HR, Lactate, and RPE of each cadence. VO2 max, HR, lactate and RPE evidenced significant differences among cadences in the 8-minute test. The cadence of 45 strokes·min-1 presented a significant lower VO2 max, HR, and lactate than 55 strokes·min-1 and 65 strokes·min-1 in the 8-minute test. There were also significant differences lower lactate and RPE values in 55 strokes·min-1 than 65 strokes·min-1 in the 8-minute test.  On the other hand, HR and RPE was higher at 8 minutes than 4 minutes in 55 strokes·min-1 and 65 strokes·min-1. However, at minute 8 lactate presented a significant higher value than at minute 4 in 65 strokes·min-1.”

Discussion

REVIEWER: line 235:withdraw to the best of our knowledge

AUTHORS: Thank you for your observation. We have deleted it.

REVIEWER: line 252-253: conversion of type 2 to  ... authors have not assessed this data. Withdraw

AUTHORS: Thank you for your recommendation. The authors have deleted it.

REVIEWER: line 285: to allow for...

AUTHORS: Thank you for your recommendation. The authors have solved it.

Reviewer 2 Report

A generally well written manuscript that presents some interesting data. Some issues need to be addressed.

Results

Table 2. Interaction was shown for efficiency, but t-test did not shown difference between values in minute 4 and minute 8. In addition, one-way ANOVA was not observed significant effect of cadence for efficiency in minute 8, but post-hoc test indicated difference between cadences. I consider that these conflicts are due to difference of applied test statistics. How do you interpret this result?

To avoid statistical contradiction, I prefer to avoid two-way ANOVA analysis in this study and using post-hoc test only when one-way ANOVA shows significant difference.

L185-187, Table 2. Is there significant difference for RER in 55spf between at minute 8 and minute 4? Is so, please add statistical symbol in Table 2. Additionally, number of significant figures of RER at 45spf should be revised.

Table3. What is the difference between statistical symbol “a” and “b”? Please check captions and symbol carefully.

Discussion

L257-L265. It is difficult to follow why difference in VO2 kinetics and muscle fiber type between arm and leg are involved with change in paddling efficiency depending on the cadence. Could you explain more clearly?

In this context, I presumed that lower efficiency at high cadence is partly due to extra energy requirement for postural keeping and/or increase in ATP consumption for ionic transport with a single contraction. How do you interpret about these factors? 

L270-272. If you calculate percentage of fat oxidation, you should describe calculation process in Material and Method section.

Conclusion

I prefer to specify that result was obtained in laboratory condition. Showing applicability limit is meaningful for athletes and coaches since SUP is generally conducted in the field as authors suggested.

Author Response

Point-by-Point Response to Reviewer’s Comments

We would like to sincerely thank the reviewers for their helpful recommendations. We have seriously considered all the comments and carefully revised the manuscript accordingly. Revisions are highlighted in yellow through the manuscript to indicate where changes have taken place. We feel that the quality of the manuscript has been significantly improved with these modifications and improvements based on the reviewers’ suggestions and comments. We hope our revision will lead to an acceptance of our manuscript for publication in International Journal of Environmental Research and Public Health.

In advance,

King regards

REVIEW 2

A generally well written manuscript that presents some interesting data. Some issues need to be addressed.

Results

REVIEWER: Table 2. Interaction was shown for efficiency, but t-test did not shown difference between values in minute 4 and minute 8. In addition, one-way ANOVA was not observed significant effect of cadence for efficiency in minute 8, but post-hoc test indicated difference between cadences. I consider that these conflicts are due to difference of applied test statistics. How do you interpret this result?

To avoid statistical contradiction, I prefer to avoid two-way ANOVA analysis in this study and using post-hoc test only when one-way ANOVA shows significant difference.

AUTHORS: Thank you for your observation. To avoid misunderstandings and following your recommendation we have eliminated the values of the 2-way ANOVA. We have left only the 1-way ANOVA.

REVIEWER: L185-187, Table 2. Is there significant difference for RER in 55spf between at minute 8 and minute 4? Is so, please add statistical symbol in Table 2. Additionally, number of significant figures of RER at 45spf should be revised.

AUTHORS: Thank you for your recommendation. The authors have reviewed results based on the data in the tables.

REVIEWER: Table3. What is the difference between statistical symbol “a” and “b”? Please check captions and symbol carefully.

AUTHORS: Thank you for your observation. We have included “a” and “b” meaning: a: Significant differences regarding 45spm using Bonferroni test. b: Significant differences regarding 55spm using Bonferroni test.

Discussion

REVIEWER: L257-L265. It is difficult to follow why difference in VO2 kinetics and muscle fiber type between arm and leg are involved with change in paddling efficiency depending on the cadence. Could you explain more clearly?

AUTHORS: Thank you for your appreciation. We have simplified the paragraph by trying to summarise and set out the ideas more clearly.

REVIEWER: In this context, I presumed that lower efficiency at high cadence is partly due to extra energy requirement for postural keeping and/or increase in ATP consumption for ionic transport with a single contraction. How do you interpret about these factors?

AUTHORS: We thank the reviewer for his insightful comment, he makes an interesting point. Previous studies have suggested that the efficiency of fast fibers is slightly lower than that of slow fibers (Woledge, 1968; Gibbs and Gibson, 1972; Wendt and Gibbs, 1973; Reggiani et al., 1997), which would partially explain why higher cadences displayed worse efficiency values in our study.  However, since other researchers have found the opposite (Heglund and Cavagna, 1987) we cannot ensure that this was the main/only reason. Within other possible alternative explanations, higher cadences may be related to greater instability, which would imply higher needs of muscle activation for postural control and would ultimately lead to a higher energy consumption, and therefore, worse efficiency. However, muscle activation was not measured in our study and we cannot confirm that this was the case.

REVIEWER: L270-272. If you calculate percentage of fat oxidation, you should describe calculation process in Material and Method section.

AUTHORS: Thank you for your appreciation. We have added a sentence referring to the way we have obtained the percentage of fat oxidation. “The RER results and the tables provided by Peronnet F et al. [34] were used to calculate the percentage of oxidation of the fats of each paddler in each of the cadences.”

Conclusion

REVIEWER: I prefer to specify that result was obtained in laboratory condition. Showing applicability limit is meaningful for athletes and coaches since SUP is generally conducted in the field as authors suggested.

AUTHORS: Thank you for your recommendation. We have modified the conclusion by making it clear that these data are laboratory measurements: “International male SUP paddlers were most gross efficiency and economical when paddling at 45 strokes·min-1 vs. 55 or 65 strokes·min-1, as confirmed by lower RPE values in measurements made in a laboratory. Similarly, this gross efficiency and economy shown at 45 strokes·min-1 implied a greater use of fat as an energy substrate. Those improvements may likely translate to faster paddling speed and greater endurance, and they may be helpful to coaches and athletes in determining optimal GE and economy, as these differences in competition are likely to yield meaningful improvement in performance.”

Round 2

Reviewer 2 Report

Thank you for adjusting the manuscript to the recommendations given.

Major point

>Previous studies have suggested that the efficiency of fast fibers is slightly lower than that of slow fibers (Woledge, 1968; Gibbs and Gibson, 1972; Wendt and Gibbs, 1973; Reggiani et al., 1997), which would partially explain why higher cadences displayed worse efficiency values in our study.  However, since other researchers have found the opposite (Heglund and Cavagna, 1987) we cannot ensure that this was the main/only reason. Within other possible alternative explanations, higher cadences may be related to greater instability, which would imply higher needs of muscle activation for postural control and would ultimately lead to a higher energy consumption, and therefore, worse efficiency. However, muscle activation was not measured in our study and we cannot confirm that this was the case.

Thank you for your comment. I can understand your opinions. But then, how do you explain physiological reason for lower efficiency at high cadence in SUP? Difference in efficiency between cadence is major finding of your study, and you should discuss this point in this paper.

Minor point

Table2, 3. Please delete caption about two-way ANOVA.